# Design of a Four-Port Flexible UWB-MIMO Antenna with High Isolation for Wearable and IoT Applications

**DOI:** 10.3390/mi13122141

**Published:** 2022-12-03

**Authors:** Jie Zhang, Chengzhu Du, Ruohui Wang

**Affiliations:** College of Electronics and Information Engineering, Shanghai University of Electric Power, Shanghai 200090, China

**Keywords:** CPW, MIMO, IoT, flexible antenna, wearable, liquid crystal polymer (LCP)

## Abstract

A 2 × 2 ultra-wideband MIMO flexible antenna with a low profile and good isolation was designed for Internet of Things (IoT) realms and wearable devices. The antenna elements were placed on a novel flexible substrate of liquid crystal polymer (LCP) with compact dimensions fed by a coplanar waveguide (CPW). In order to ameliorate isolation, the cross-shaped decoupling branches were placed among the antenna elements. The proposed UWB antenna can operate from 2.9 GHz to 10.86 GHz with a good reflection coefficient of S_11_ < −10 dB as well as a high isolation better than 22 dB. Its operating bands include 5G, WiFi, X-band, etc. Moreover, the parameters of diversity performance were also tested. These parameters included an average gain of approximately 4 dB, a low ECC of less than 0.01, and good diversity gain of 9.999. The flexible MIMO antenna performs well in bending and on-body conditions. To sum up, the antenna has good prospects in IoT applications and wearable fields.

## 1. Introduction

Ultra-wideband (UWB) technology is the most promising solution to the scarcity of spectrum resources, ranging from 3.1 to 10.6 GHz [1]. UWB technology possesses large system capacity, high communication rates and low system complexity [2]. In the present era, many scholars have studied UWB [3,4]. Combining MIMO with UWB can improve RF bandwidth availability and help deal with multi-path fading problems and co-channel interference. Several articles about ultrawideband MIMO antennas have been published in recent years [5,6,7,8,9,10,11].

With the growing number of intelligent devices requiring internet access, there is an increasing requirement for Internet of Things (IoT) technology. The study of IoT antennas has become a popular choice among scholars. In [12], an elliptical patch antenna inverted on a T-shape stub had a broadband peculiarity (3.8–12 GHz) for IoT fabricated on an FR4 substrate. In [13], a monopole antenna was presented with triple band-notched peculiarities for IoT fields. In [14], a UWB antenna applied to IoT devices with FR4 and denim textile substrate was presented, operating at 2.9–11 GHz. A flexible antenna with a bandwidth of 98.4% (0.856–2.513 GHz) for IoT applications was introduced in [15]. In [16], a circular-shaped flexible antenna inverted on an inside-cut feeding structure was fabricated on a bendable PET substrate operating at 3.04–10.7 GHz and 15.18–18 GHz.

In order to improve spectral efficiency and increase communication quality, many MIMO antennas for IoT applications have been designed over the past few years. A triband MIMO antenna based on FR4 was presented in [17], covering multiple wireless bands, especially the 1900 MHz band supporting NB-IoT. In [18], an eight-port UWB-MIMO antenna operating at 3–12 GHz was proposed for IoT applications based on Metallic Via. In [19], a multi-port MIMO antenna with adjustable ports and regulatable working frequency was presented, printed on FR4 tailored for IoT devices based on WLAN and UWB. When used in wearable and IoT applications, antennas are often embedded in clothing or irregularly shaped devices. Therefore, they are usually made from flexible materials in order to ensure that they can work effectively without affecting the operation of equipment or people’s everyday activities. In [20], a 2-port antenna functioned in IoT devices that is capable of operating at 3.1–13 GHz and offers an excellent isolation level of −23 dB was printed on Kapton Polyimide. In [21], a wideband MIMO antenna was demonstrated, integrating with DGS based on jeans working in UWB and WLAN with excellent isolation of 32 dB. In [22], a 2-port UWB antenna fabricated on a Kapton polyimide flexible material was introduced with enhanced isolation over 15 dB. An antenna printed on a bendable substrate (FR-4) that can operate at 3.89–17.09 GHz with a stable gain was proposed in [23]. A compact MIMO antenna comprising four crescent monopoles was introduced in [24], printed on a Roger 3003 substrate with an impedance bandwidth of 3.1–11 GHz.

This paper presents a CPW-fed flexible four-element UWB-MIMO antenna for wearable and IoT applications. The four-element MIMO antenna is printed on a liquid crystal polymer (LCP) substrate, which has the advantages of flexibility, low dielectric loss, robustness, and high thermal endurance. The cross-shaped branches are inserted between element antennas in this design for good mutual decoupling. The four-port ultrawideband MIMO antenna has good performance with wider impedance bandwidth (S_11_ < −10 dB) of 2.9–10.86 GHz, high isolation lower than −22 dB and ECC < 0.01. Moreover, the designed antenna shows the following advantages: (1) the antenna is integrated easily with CPW feed; (2) can work at ultrawideband frequencies; and (3) is based on an ultrathin flexible LCP substrate that is appropriate for wearable and IoT applications.

## 2. Antenna Design

### 2.1. Single Element Antenna

In Figure 1a, the initial antenna geometry is described. It is fed by a coplanar waveguide and printed on a 0.1 mm thick LCP substrate with a relative dielectric constant = 2.9 and loss tangent = 0.002. Two arc-shaped corners are cut off on the ground plane for good impedance matching. Figure 1b shows a depiction of the simulated result of the antenna’s S_11_. The operating range of the antenna introduced in this article is 2.66–11.39 GHz. Antenna parameters after optimization are given in Table 1.

### 2.2. Four-Element Flexible UWB Antenna

A four-port flexible MIMO antenna array was designed with a cross-shaped branch inserted between orthogonal element antennas for high isolation. Figure 2 illustrates the geometry of the proposed antenna with a small size of 65 mm × 65 mm.

### 2.3. Isolation Structure Analysis

A diagram of the decoupling structure design process (from Ant-1 to Ant-4) is exhibited in Figure 3. Ant-1 shows that four-element antennas are placed orthogonally with each other without a branch isolator. On the base of Ant-1, Ant-2 was designed with a branch isolator, Ant-3 with a two-branch isolator, and Ant-4 with a three-branch isolator, respectively. Due to orthogonality, the perpendicular arrangement of the antenna elements improves the isolation to a certain extent.

To further comprehend the mechanism of the branch isolator, S-parameters of four antennas are presented in Figure 4. Due to the symmetry of these four elements, the verdict of S_12_ = S_14_ = S_23_ = S_34_ = S_43_ = S_32_ = S_41_ = S_21_ and S_13_ = S_24_ = S_42_ = S_31_ can be determined. For brevity, only the results of S_11_, S_21_ and S_31_ are compared in Figure 4. With an increasing number of decoupling branches, the S_11_ values of the four antennas change slightly. In Ant-1, the values of S_21_ are all lower than −20 dB in the operational frequency band, but the values of S_31_ are greater than −16 dB in the bands of 2.7–4 GHz. In order to degrade S_31_, the number of branches was increased from 1 to 3, and the values of S_31_ were gradually lower than −20 dB in all working bands. In the operating bands (2.8–11 GHz), the S_21_ values of the four antennas were all lower than −20 dB, and the S_31_ values of Ant-4 were better than those of Ant-3 in most cases; ultimately, Ant-4 was the best choice.

In Figure 5, the current distributions of the MIMO antenna with and without isolation branch at 6 GHz are shown. It can be seen that when the MIMO antenna has an isolation branch added, antenna element 3 at the lower left corner in Figure 5b has a lower surface current than the one in Figure 5a. It can be inferred that the cross-shaped branches degrade the coupling of diagonally placed elements.

## 3. Simulated and Measured Results

### 3.1. S-Parameter

In Figure 6, both simulated and measured curves are compared. Due to errors of processing and loss of the antenna test connector, the measured S_11_, S_22_ and S_33_ were affected slightly. Additionally, the measured curves of isolation were slightly better than the simulated ones, which were all less than −22 dB across the entire working bands. According to the above conclusions, the simulation and test results for the antenna met the design requirements.

### 3.2. Radiation Patterns

Figure 7 shows 2-D radiation patterns of the proposed four-port flexible antenna on the E-plane and H-plane. The E-plane refers to the YOZ plane, which parallels the direction of the electric field. The H-plane refers to the XOZ plane, which parallels the direction of the magnetic field. The simulation and test results were obtained when port 1 was used as the excitation, and the other ports were matched with 50 Ω loads. On the E-plane, there were steady bidirectional radiation patterns that resembled the number "8" at 4 GHz and 6 GHz, but they were slightly distorted at 8 GHz. Additionally, the radiation patterns presented omnidirectional characteristics on the H-plane. 

## 4. Diversity Performance

### 4.1. ECC and DG of the Flexible MIMO Antenna

The envelope correlation coefficient (ECC) describes the degree of correlation between antenna elements in a MIMO system, which can be estimated using Equation (1) with S-parameters.
(1)$$\rho_{e}\left(i,j,N\right)={\left|\frac{\sum_{n=1}^{N}S_{i,n}^{*}S_{n,j}}{{\left|\prod_{k=i,j}\left(1-\sum_{n=1}^{N}S_{k,n}^{*}S_{n,k}\right)\right|}^{\frac{1}{2}}}\right|}^{2}$$

The antenna must have relatively high efficiency to maintain the validity of Equation (1). By using the following Equation (2), then, we can also calculate the ECC more accurately based on the far-field radiation of the antenna [25]:(2)$$\rho_{e}\left(i,j\right)=\frac{{\left|\iint_{4\pi }^{\;}{\vec{F}}_{i}\left(\theta ,\varphi \right)\cdot {\vec{F}}_{j}\left(\theta ,\varphi \right)d\Omega \right|}^{2}}{\iint_{4\pi }^{\;}{\left|{\vec{F}}_{i}\left(\theta ,\varphi \right)\right|}^{2}d\Omega \cdot \iint_{4\pi }^{\;}{\left|{\vec{F}}_{j}\left(\theta ,\varphi \right)\right|}^{2}d\Omega }$$

Diversity gain (DG) shows the loss of transmission power on condition that the diversity mechanism is implemented in a multiple-input multiple-output system. When the diversity mechanism is applied to a MIMO system, the diversity gain (DG) is the loss of transmission power. Therefore, DG is also a crucial characteristic of the antenna, and it can be derived with Equation (3).
(3)$$\mathrm{DG}=10\sqrt{1-{\left(\mathrm{ECC}\right)}^{2}}$$

An efficient MIMO antenna should meet ECC < 0.5 and DG around 10. For brevity, Figure 8 and Figure 9 only show the relevant parameters between antenna port 1 and port 2. A comparison of tested and simulated ECC curves is depicted in Figure 8; all ECC values are lower than 0.01 in the working band. As Figure 9 illustrates, the values of DG are all above 9.999. In summary, the proposed antenna has good diversity.

### 4.2. Gain and Radiation Efficiency

Figure 10 depicts the UWB antenna gain after simulation and measurement. Due to uncertainties associated with the testing environment, the measured curve was generally lower than the simulated one. The measured gain changed from 0.95 to 6.49 dB, and its average value was about 4 dB. Figure 11 demonstrates that the antenna’s efficiency was greater than 85% across the entire band, indicating that this proposed antenna has low power consumption.

### 4.3. Total Active Reflection Coefficient (TARC)

In MIMO antenna systems, TARC is a significant performance indicator of total incident power [26]. It can be calculated with Equation (4):(4)$$\mathrm{TARC}=N^{-0.5}\sqrt{\sum_{i=1}^{N}{\left|\sum_{k=1}^{N}S_{\mathrm{ik}}e^{{j\theta }_{k-1}}\right|}^{2}}$$

Figure 12 depicts the measured values of TARC in different degrees. As can be seen from the curves, TARC values within the working band were all lower than −10 dB, indicating that the antenna presents good radiation characteristics.

## 5. Flexible Study

### 5.1. Flexibility Analysis

Flexible antennas are often used on wearable devices worn on people’s arms. The radius of the human forearm is about 30 mm, and the radius of the primary arm is about 40 mm. Therefore, the S-parameters of this presented four-port flexible antenna were tested when the antenna was fixed on various cylinders (R = 30 mm and R = 40 mm) on the E-plane or H-plane, respectively. The results of the measurements are exhibited in Figure 13. In the high frequencies, when the antenna is bent along the E-plane, its bandwidth is affected more than the bending along the H-plane. In different bending cases, the S_11_ values of the antenna all meet the actual working requirements. It was observed that the curves of S_21_ and S_31_ in different cases were less than −20 dB, indicating that the antenna shows good isolation. Moreover, the performance of the antenna bending along the H- plane was better than that along the E-plane.

### 5.2. Analysis of On-Body Condition

Since the proposed flexible UWB antenna is an ideal candidate for IoT and wearable fields, the S-parameters of this UWB-MIMO antenna, when applied to the human body, were analyzed and researched. The tested results of scattering parameters when the antenna was placed on the arm, leg and back are depicted in the following Figure 14. The curves of S_11_ were slightly distorted when the flexible antenna was placed on a person. Although the working bandwidth of the antenna narrowed, it was still in compliance with the working requirements. Furthermore, the measured results for S_21_ and S_31_ were all lower than −22 dB, demonstrating the high isolation of the designed antenna.

The specific absorption rate (SAR) is a vital parameter for quantifying antenna radiation’s effects on humans [27]. In Figure 15, a human body tissue model is used to simulate SAR by HFSS, which was made up of a 2 mm skin layer, 5 mm fat layer, 20 mm muscle layer, and 13 mm bone layer. Parameter H represents the distance from the tissue model to the antenna. Table 2 lists the maximum SAR values of the presented antenna attained in various conditions. The simulated values all meet the European Union Standard (2 W/kg/10 g).

## 6. Performance Comparison

Table 3 shows a comparison between the proposed flexible ultra-wideband four-element MIMO antenna and previous designs. Antennas in [10,11,18] were 4-port and 8-port compact antennas with good performance. In comparison with the presented antenna, they were printed on rigid FR4 with microstrip feed, which is unsuitable for wearable applications and integration. In [15,16,20,21,22,23,24], the UWB antennas were all designed for IoT applications. The antennas in Refs. [15,16] were only single wideband antennas, and the antennas in Refs. [20,21,22] were two-port MIMO antennas. As a result, their transmission capacity was less than that of the proposed antenna. The isolation of the proposed antenna is higher than those in [23,24]. Compared with the antennas discussed above, the proposed CPW-fed four-port MIMO antenna has better isolation, wider bandwidth, higher efficiency and easier integration.

## 7. Conclusions

An innovative four-port flexible ultra-wideband MIMO antenna based on LCP substrate was designed and presented in this paper. The MIMO antenna consists of four antenna elements placed orthogonally and cross-shaped decoupling branches for high isolation. The antenna has a low profile in size (65 × 65 × 0.1 mm^3^). The measured impedance bandwidth (S_11_ < −10 dB) ranges from 2.9 GHz to 10.86 GHz. A high level of isolation between ports was observed in the measurements of over 22 dB. The designed flexible MIMO antenna has good diversity performance with a low ECC (<0.01) and high DG (>9.999). Moreover, the presented antenna performs well in bending and on-body conditions. Accordingly, it is an appropriate candidate for wearable applications and IoT fields.

## Figures and Tables

**Figure 1 micromachines-13-02141-f001:**
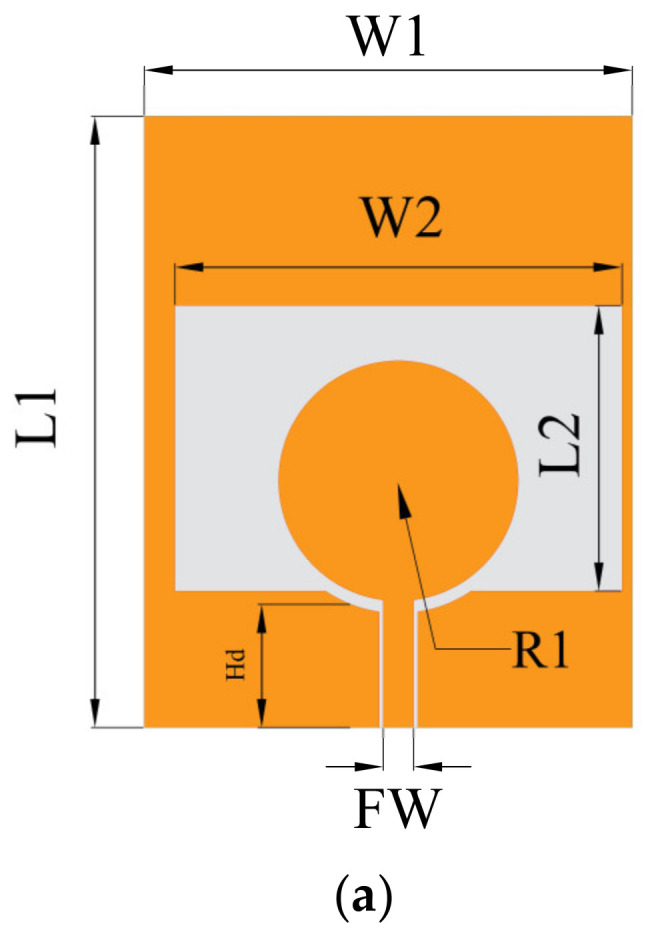
Single−slot antenna. (**a**) Geometrical details, (**b**) simulation results of S_11_.

**Figure 2 micromachines-13-02141-f002:**
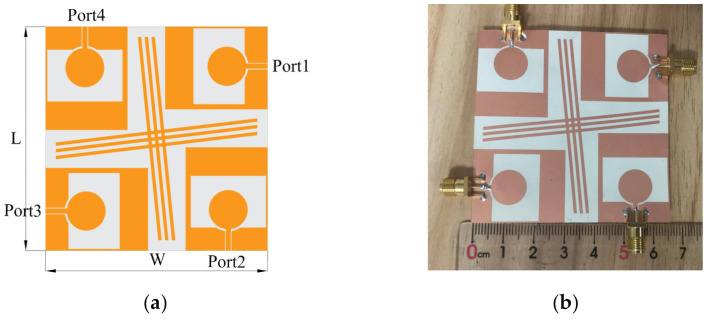
Geometry and photo of the presented antenna. (**a**) Geometrical details, (**b**) fabricated antenna.

**Figure 3 micromachines-13-02141-f003:**
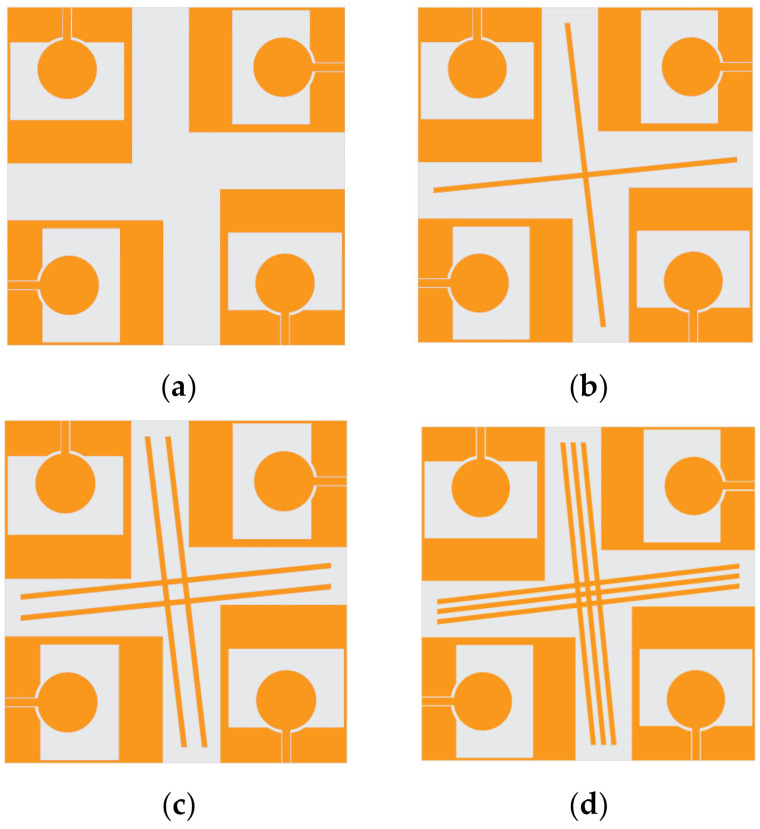
Evolution of the MIMO antenna (**a**) Ant-1, (**b**) Ant-2, (**c**) Ant-3 and (**d**) Ant-4.

**Figure 4 micromachines-13-02141-f004:**
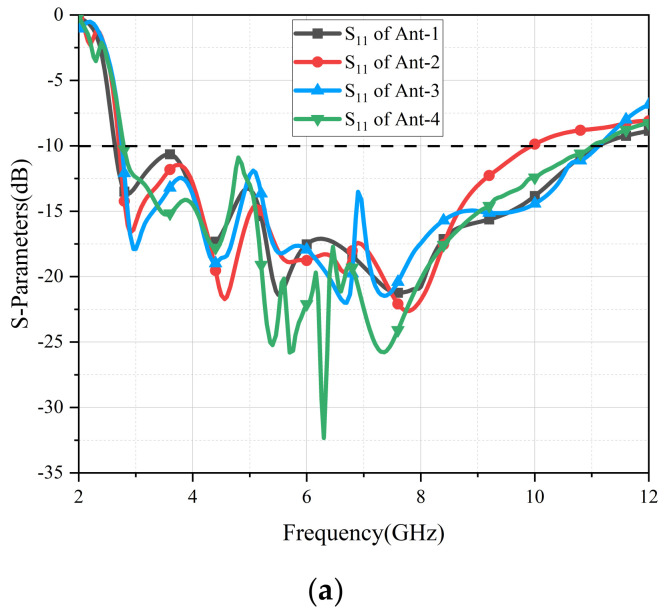
S−parameters of four antennas. (**a**) S_11_, (**b**) S_21_, (**c**) S_31_.

**Figure 5 micromachines-13-02141-f005:**
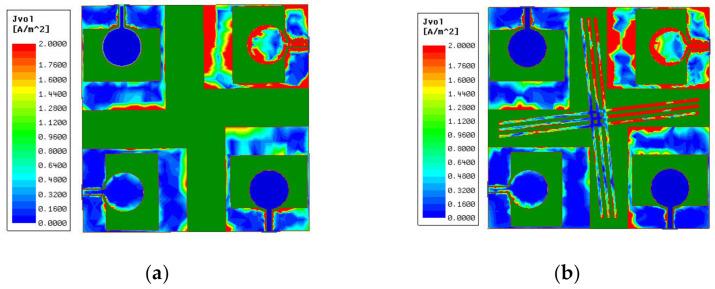
Simulated current distributions. (**a**) Without isolation branch, (**b**) with isolation branch.

**Figure 6 micromachines-13-02141-f006:**
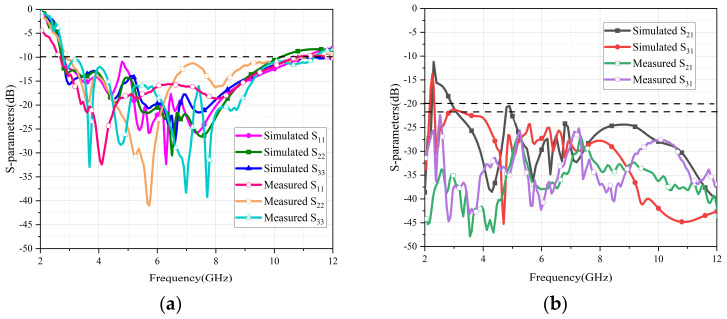
Simulation and evaluation of antenna S−parameters. (**a**) Return loss of the antenna, (**b**) isolation of the antenna.

**Figure 7 micromachines-13-02141-f007:**
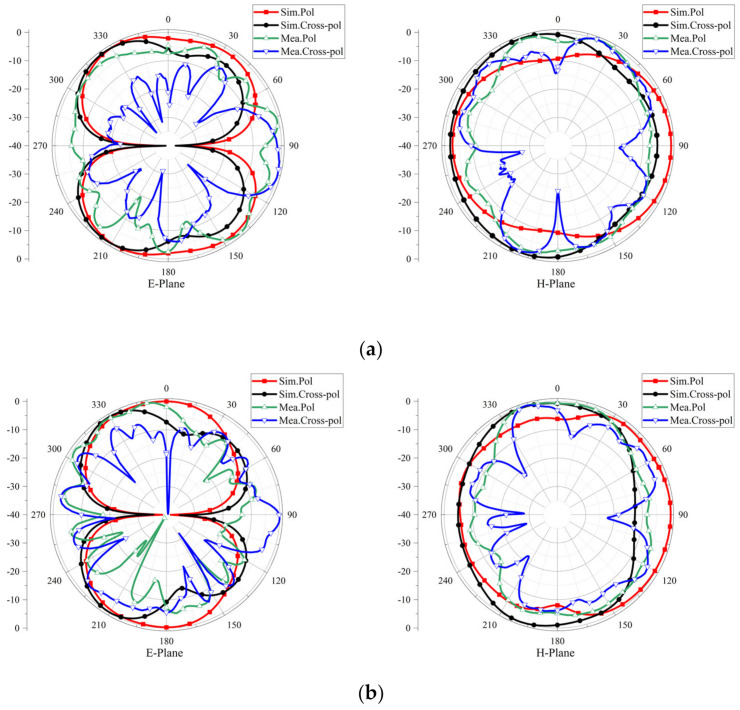
Radiation patterns of flexible four−port MIMO antenna at (**a**) 4 GHz, (**b**) 6 GHz and (**c**) 8 GHz.

**Figure 8 micromachines-13-02141-f008:**
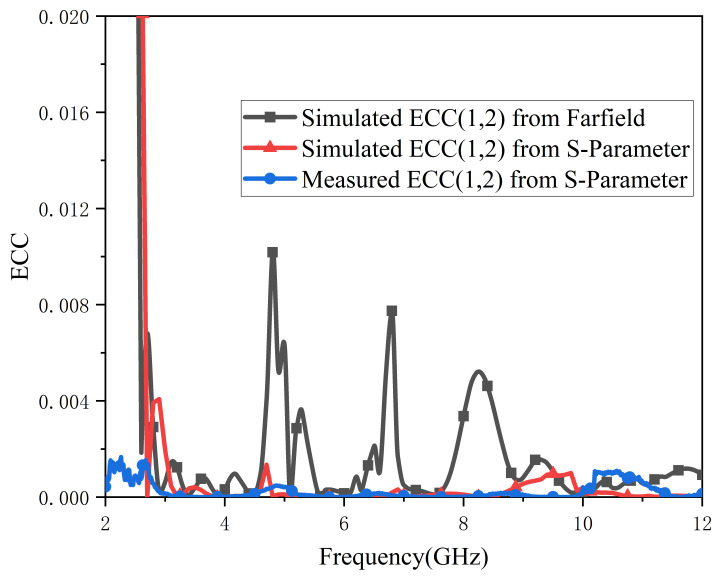
ECC (1,2) of proposed antenna.

**Figure 9 micromachines-13-02141-f009:**
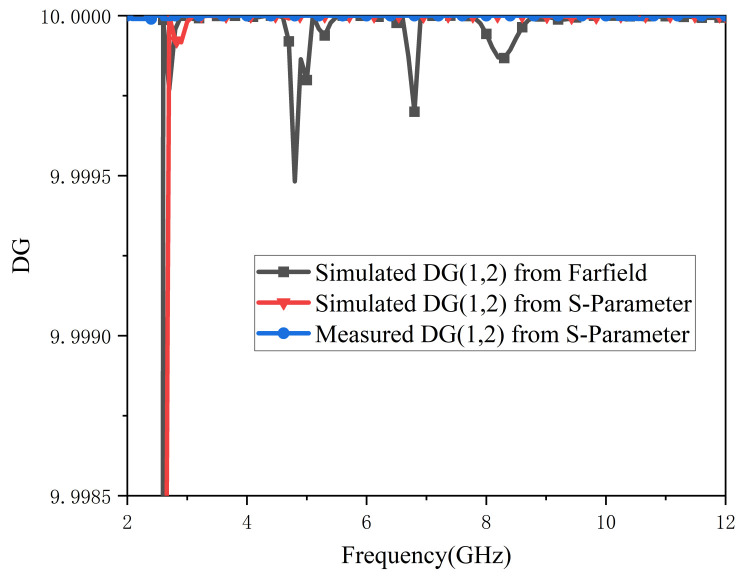
DG (1,2) of proposed antenna.

**Figure 10 micromachines-13-02141-f010:**
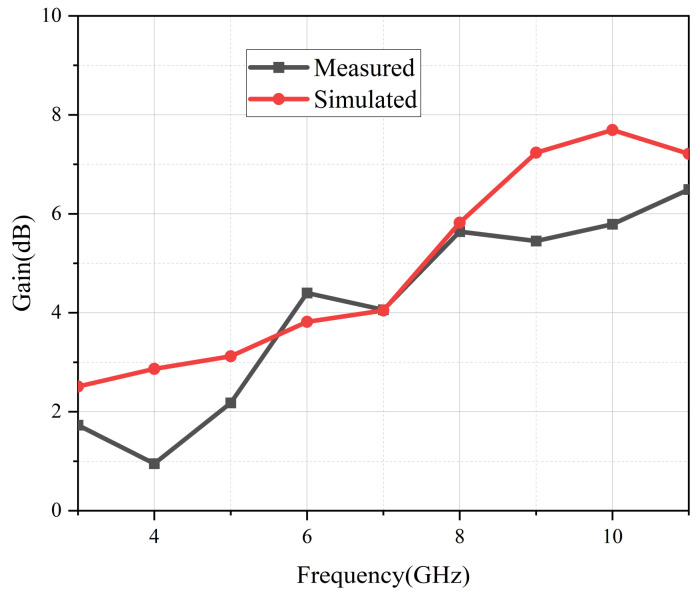
Simulated and measured gain.

**Figure 11 micromachines-13-02141-f011:**
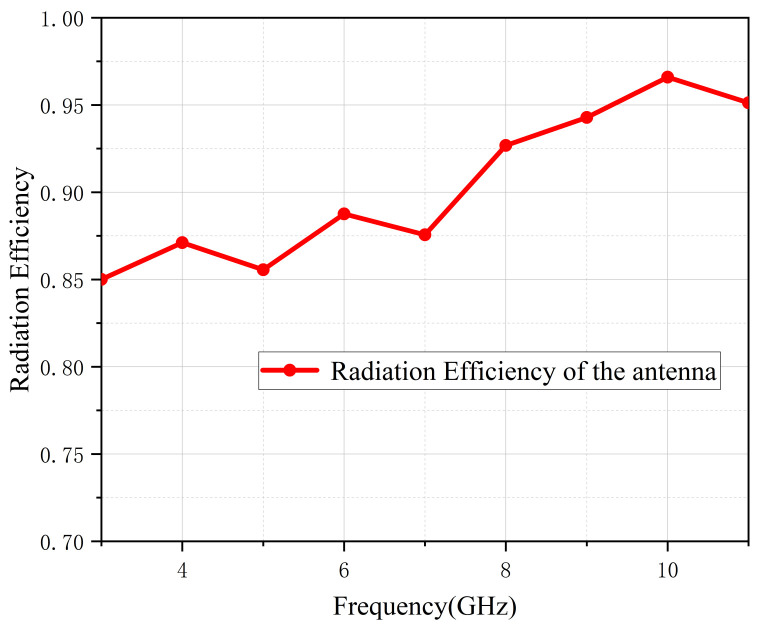
Radiation efficiency.

**Figure 12 micromachines-13-02141-f012:**
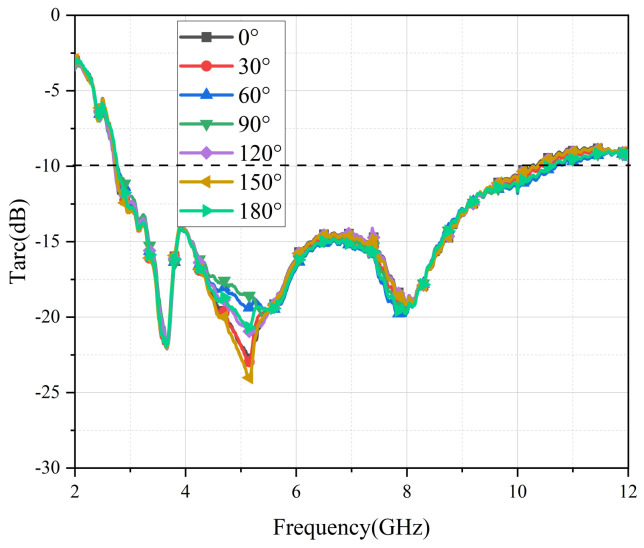
Measured curves of TARC in different degrees.

**Figure 13 micromachines-13-02141-f013:**
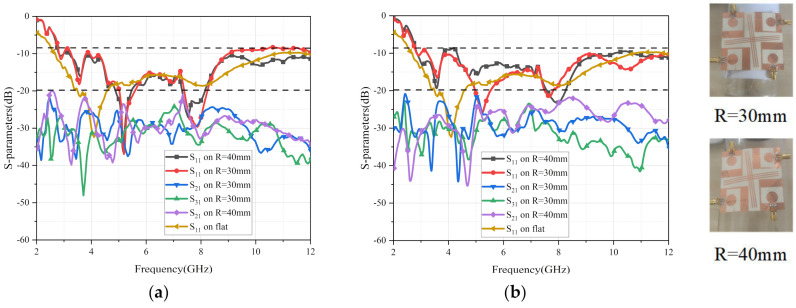
S−parameters in different conditions. (**a**) E−plane, (**b**) H−plane.

**Figure 14 micromachines-13-02141-f014:**
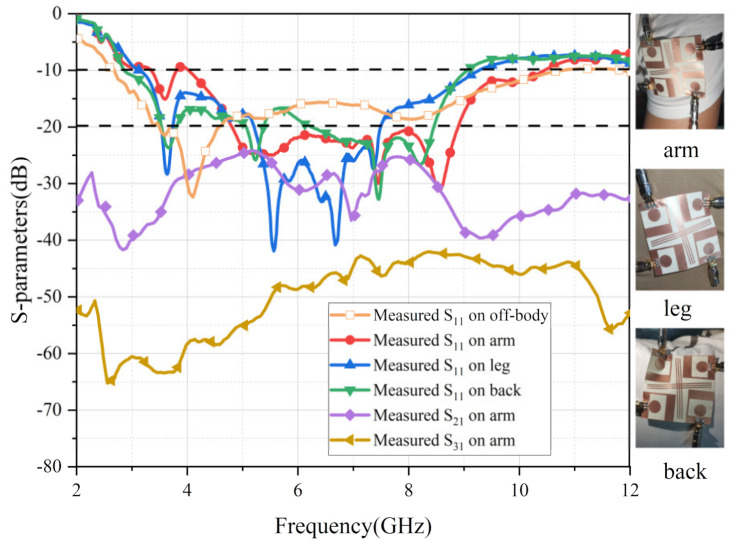
The tested performance on a person’s body.

**Figure 15 micromachines-13-02141-f015:**
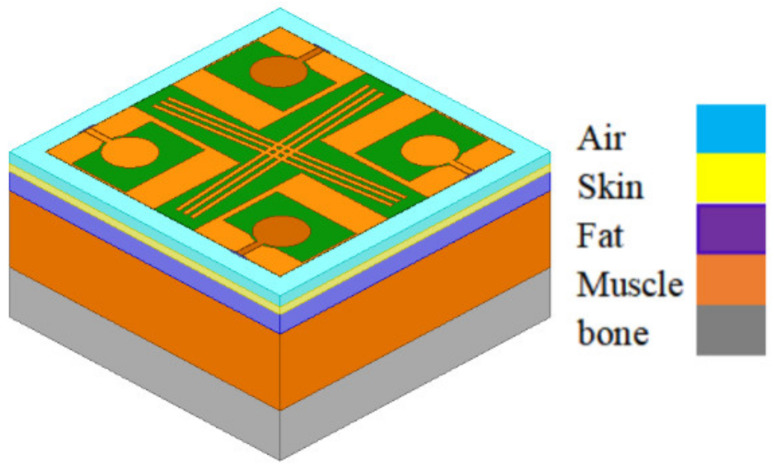
Model of the four-port MIMO antenna on human body tissue.

**Table 1 micromachines-13-02141-t001:** Antenna dimensions (Unit: mm).

L1	L2	FW	R1	W1	W2	Hd
30	12	1.56	5.9	22	9.5	8

**Table 2 micromachines-13-02141-t002:** Maximum SAR values under various conditions.

Frequency	4 GHz	6 GHz	8 GHz
10 g Tissue (W/kg) (H =3 mm)	1.04	0.59	0.89
10 g Tissue (W/kg) (H =5 mm)	0.86	0.45	0.79

**Table 3 micromachines-13-02141-t003:** Comparison between the designed antenna and others.

Ref.	Substrate	Bands (GHz)	Size (mm^3^)	Isolation (dB)	Port Number	ECC	DG	Feeder Method
[10]	FR4	3.1–10.8	24 × 24 × 0.8	<−20	4	<0.008	NA	Microstrip feed
[11]	FR4	2.94–14	40 × 40 × 1.575	<−17	4	<0.03	NA	Microstrip feed
[15]	Rubber	0.856–2.513	92 × 121.5 × 2	NA	1	NA	NA	Microstrip feed
[16]	PET	3.04–10.7 15.18–18	47 × 25 × 0.135	NA	1	NA	NA	CPW feed
[18]	FR4	3–12	54 × 54 × 1.6	<−20	8	<0.009	9.9	Microstrip feed
[20]	Kapton polyimide	3.1–13	100 × 22 × 0.508	<−23	2	<0.01	9.98	CPW feed
[21]	Jeans textile	3.1–10.6	30 × 50 × 1.5	<−32	2	<0.12	9.8	Microstrip feed
[22]	Kapton polyimide	2.9–12	22 × 33 × 0.125	<−15	2	<0.3	NA	Microstrip feed
[23]	FR4	3.89–17.09	56 × 68 × 0.2	<−15	4	<0.02	NA	Microstrip feed
[24]	Roger3003	3.5–11	54 × 54 × 0.13	<−17	4	<0.03	NA	CPW feed
Prop.	LCP	2.9–10.86	65 × 65 × 0.1	<−22	4	<0.01	9.999	CPW feed

## Data Availability

Some or all data, models, or code generated or used during the study are available from the corresponding author by request.

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
