# Peer review of "Design of a Four-Port Flexible UWB-MIMO Antenna with High Isolation for Wearable and IoT Applications"

_micromachines, 2022, doi:10.3390/mi13122141_

Round 1

Reviewer 1 Report

In this paper, a coplanar waveguide-fed flexible four-element UBW-MIMO antenna for wearable and IoT applications is introduced, which has the advantages of flexibility, low dielectric loss, robustness, and high thermal endurance. There are some questions as follow.

 1. In the proposed four element UWB antenna architecture, the position of each antenna is fixed. The four-port flexible MIMO antenna array is designed with a cross-shaped branch inserted between orthogonal element antennas, the good isolation performance can be understood through isolation structure analysis. However, how does this fixed structure reflect flexibility? More explanation for the advantages of flexibility should be given in the face of this fixed structure.

 2. Compared with the exist antennas, the proposed CPW-fed four-port MIMO antenna has better isolation, wider bandwidth and easier integration. However, whether the cost and power consumption of the proposed antenna have advantages? The authors need to give a more detailed comparison of cost and power consumption.

 3. In the simulated and measured results analysis, only the comparison between the simulation results and the measured results is available, and the simulation or measured comparison between the proposed CPW-fed four-port MIMO antenna and other existing antennas is lacking.

 4. The advantages of the proposed antenna are brought by the antenna structure or the use of isolation branch? Whether the material of the isolation branch will affect the performance of the proposed antenna.

 5. Please draw all curves more smooth. In addition, please improve the resolution of all figures.

 6. Please check and correct some typos and grammar errors.

Reviewer 2 Report

This paper presents UWB MIMO antenna based on the LCP substrate.

The antenna topology is a well-known structure. Therefore, the main novelty of this work relies on the decoupling structure. The detailed operating principle of the decoupling mechanism, however, was not presented in the submitted manuscript. For example, the effect of the decoupling structure, branch isolator, is not clear from Fig. 4 (b). I think the isolation property mainly comes from the perpendicular arrangement. It is recommended to supplement the decoupling section.

There is a minor comment.

- There are mistype on page 4. S21, S21 --> S11, S21.

Reviewer 3 Report

This paper presents an interesting study of a flexible UWB MIMO antenna for wearable applications. The proposed antenna is fabricated and measured. I would like to give some comments to improve the quality of the manuscript.

1.      The first sentence in the first paragraph of Section I is advised to be deleted.

2.      The word of “the introduced antenna” is advised to be replaced with “the proposed antenna”.

3.      In the last paragraph of page 5, the sentence of “Following Figure 7 exhibits 2-D radiation patterns of the introduced four-port flexible antenna at E-plane and H-plane.” is advised to be revised to “Figure 7 shows 2-D radiation patterns of the proposed four-port flexible antenna at E-plane and H-plane”.

4.      For Figures 1-7,12, the titles of the sub-figures are given twice. I think once if OK.

Reviewer 4 Report

Dear authors,

Please find below my comments on the manuscript id: micromachines-1997963, entitled "A design of four-port flexible UWB-MIMO antenna with high isolation for wearable and IoT applications".

The article describes a 2×2 sequentially rotated arrangement of monopoles printed on a flexible substrate. The antenna array is intended for UWB MIMO in wearable and IoT applications. The antenna presents good port match and isolation from 2.9 GHz to 10.9 GHz. A cross-shaped decoupling structure is used among antenna elements to increase isolation. An average gain of 4 dBi is presented, as well as an ECC below 0.002. Tests reporting the effects of flexing the substrate on the port parameters are also shown.

The authors claim that the main contributions of the article are the compact antenna size, the use of a CPW feed, the port match and inter-port isolation in the UWB frequency range, and the use of a flexible substrate of liquid crystal polymer (LCP). Nevertheless, the claimed novelty of the presented antenna architecture is questionable, as it is similar to previously reported structures which are not properly acknowledged in the manuscript. Furthermore, the advantages and unique features of the proposed design are not well established.

Specific comments that motivate this review are:

Legends in Figure 4b are hard to read.

The last paragraph of page 4: "...Figure 5, the current distribution ... demonstrates that cross-shaped branches serve as a reflector...," is misleading. In the first place, because the mentioned figure is not suitable for demonstrating that the mentioned reflection effect is present, in second place, a reflection may be sensitive to frequency; therefore, this affirmation may not hold in the whole UWB frequency range.

The color and trace convention in figure 6b does not allow a fair comparison of results.

The results concerning the radiation pattern are not very clear as the measurement setup is not stated. In the first place, the state of the antenna ports is not mentioned. On the other hand, the E- and the H- planes of the structure are not defined.

In the last paragraph of page 5, it mentions that the measurements are taken at the same "resonant" frequencies of the antenna. However, those resonant frequencies are not previously discussed.

In figure 7, the measured pattern cuts of the co-pol and x-pol components do not seem to agree with the simulation expectations. Furthermore, x-pol seems to be even bigger than co-pol components.

Results in Figure 10, show a big difference between the measured and the expected gain (>4dB in some cases). The cause of this discrepancy must be further addressed. What is the efficiency of the antenna?

Results in Fig 12. are hard to understand, as there is not a detailed description of the different conditions employed for performing the corresponding measurements, nor the significance of these results is discoursed.

Finally, Concerning the use of references, the manuscript has some deficiencies that must be tackled.

Reference 21 is not cited (although eq. 1 reproduces the expression therein).

Table 3, presenting the comparison between the proposed design and other alternatives existing in the literature, seems incomplete and does not include relevant works, with very similar designs and results. Some of those works are:

• Ibrahim, A.A., Ahmed, M.I. & Ahmed, M.F. A systematic investigation of four ports MIMO antenna depending on flexible material for UWB networks. Sci Rep 12, 14351 (2022). https://doi.org/10.1038/s41598-022-18551-8

• Desai, J. Kulkarni, M. M. Kamruzzaman, Š. Hubálovský, H. -T. Hsu and A. A. Ibrahim, "Interconnected CPW Fed Flexible 4-Port MIMO Antenna for UWB, X, and Ku Band Applications," in IEEE Access, vol. 10, pp. 57641-57654, 2022, doi: 10.1109/ACCESS.2022.3179005.

• Saad, A.A.R. and Mohamed, H.A. (2019), Conceptual design of a compact four-element UWB MIMO slot antenna array. IET Microw. Antennas Propag., 13: 208-215. https://doi.org/10.1049/iet-map.2018.5163

• Srivastava, G., Mohan, A. and Chakraborty, A. (2017), A compact multidirectional UWB MIMO slot antenna with high isolation. Microw. Opt. Technol. Lett., 59: 243-248. https://doi.org/10.1002/mop.30276

Round 2

Reviewer 1 Report

No further comments.

Author Response

Thanks for your valuable comments and recognition.

Reviewer 2 Report

Thanks for the replies to the comments. However, in the vicinity of 2.1 ~ 2.3 GHz, as the number of branches increases, the isolation gets worse. (For example, isolation of ant 2 is better than ant3/4 in the corresponding band.) Nevertheless, I wonder why ant4 was selected. In particular, from the point of view of S21 (Fig. 4b), the addition of branches seems to deteriorate isolation.

Author Response

Thanks for your comments. The operating band of the UWB antenna is 3.1-10.6 GHz. The S11 of Ant-2 can only cover 2.8-10 GHz, which means the bandwidth of the antenna is narrow. The S11 values of Ant-1, Ant-3 and Ant-4 can meet the requirements of the UWB antenna (2.8-11 GHz). Therefore, we did not further optimize the antenna performance in the bands of 2.1~2.3 GHz. The S21 curves of Ant-3 and Ant-4 are both lower than - 20 dB in the operating frequency bands (2.8-11 GHz). Since the S31 and S11 values of Ant-4 are generally lower than those of Ant-3, we finally selected Ant-4.

Reviewer 4 Report

Dear authors,

I do not have new observations on the manuscript id: micromachines-1997963, entitled "A design of four-port flexible UWB-MIMO antenna with high isolation for wearable and IoT applications".

Thanks for your reply to my former comments.

Author Response

Thanks for your valuable comments and recognition. We have  polished the description and the language in this paper.